# Passive Bistatic Ground-Based Synthetic Aperture Radar: Concept, System, and Experiment Results

**Weike Feng [1],\***, **Jean-Michel Friedt [2]**, **Giovanni Nico [3]**, **Suyun Wang [1]**, **Gilles Martin [2]** **and Motoyuki Sato [4]**

[1]   Graduate School of Environmental Studies, Tohoku University, 980-8579 Sendai, Japan
[2]   FEMTO-ST, Time & Frequency Department, 25000 Besancon, France
[3]   Consiglio Nazionale delle Ricerche, Istituto per le Applicazioni del Calcolo, 70126 Bari, Italy
[4]   Center for Northeast Asian Studies, Tohoku University, 980-8576 Sendai, Japan
\*   Correspondence: feng.weike.q4@dc.tohoku.ac.jp

**Abstract:** A passive bistatic ground-based synthetic aperture radar (PB-GB-SAR) system without a dedicated transmitter has been developed by using commercial-off-the-shelf (COTS) hardware for local-area high-resolution imaging and displacement measurement purposes. Different from the frequency-modulated or frequency-stepped continuous wave signal commonly used by GB-SAR, the continuous digital TV signal broadcast by a geostationary satellite has been adopted by PB-GB-SAR. In order to increase the coherence between the reference and surveillance channels, frequency and phase synchronization of multiple low noise blocks (LNBs) has been conducted. Then, the back-projection algorithm (BPA) and the range migration algorithm (RMA) have been modified for PB-GB-SAR to get the focused SAR image. Field experiments have been carried out to validate the designed PB-GB-SAR system and the proposed methods. It has been found that different targets within 100 m (like the fence, light pole, tree, and car) can be imaged by the PB-GB-SAR system. With a metallic plate moved on a positioner, it has been observed that the displacement of the target can be estimated by PB-GB-SAR with submillimeter accuracy.

**Keywords:** ground-based synthetic aperture radar (GB-SAR); passive bistatic radar (PBR); satellite digital TV signal; synthetic aperture radar (SAR) imaging; displacement estimation

---

## 1. Introduction

Ground-based synthetic aperture radar (GB-SAR) is an important remote sensing tool for displacement measurement of open-pit mine slopes, landslides, buildings, bridges, piers, and dams with millimeter or submillimeter accuracy [1–8]. It has been viewed as an effective complement to spaceborne/air-borne SAR for imaging and displacement estimation of a local area. Compared to spaceborne/air-borne SAR, it can continuously monitor the scene of interest with a much shorter revisiting period (from several seconds to tens of minutes). Besides, its data acquisition rate can be further increased by using one-dimensional (1D) or two-dimensional (2D) multiple input multiple output (MIMO) array, which has drawn attention in the last decade [9–13]. Moreover, GB-SAR can easily obtain higher system flexibility. For example, by using a transponder based monostatic and bistatic GB-SAR system or several conventional GB-SAR/MIMO systems simultaneously, the 2D or 3D displacement vector of the target can be obtained [14–18]. Furthermore, advanced polarimetric GB-SAR imaging and interferometry techniques have also been studied [19–21]. More details and potential development directions of GB-SAR can be found in [1,3,8].

For high resolution SAR imaging, the monostatic stripmap mode is normally adopted by GB-SAR, where, to get a high azimuth resolution, the transceiver is sled along a several-meter-long rail

to generate a synthetic aperture. With respect to the transmitting signal, frequency-modulated continuous waveform (FMCW) or stepped-frequency continuous waveform (SFCW) is commonly used by GB-SAR [1]. To get a high range resolution, the frequency bandwidth of GB-SAR is typically several-hundred megahertz. The working frequency is normally within the Ku band (e.g., 17 GHz) to get a high-resolution displacement measurement. FMCW or SFCW transmitter can be achieved with an acceptable hardware requirement at the present day and GB-SAR/MIMO radar has been successfully used for various applications. However, here is a question: to further reduce its cost, improve its system flexibility, and obtain different scattering properties of the target, can we design a GB-SAR system without a dedicated transmitter, while, at the same time, its capabilities of continuous illumination, short revisiting period, high spatial resolution, and high accuracy in displacement estimation are not sacrificed? In this study, we try to answer this question by describing a passive bistatic GB-SAR (PB-GB-SAR) system, whose practical imaging and displacement estimation performance has been validated by field experiments.

The core idea of PB-GB-SAR is to exploit an existing illuminator of opportunity (IO) to replace the GB-SAR transmitter. Actually, using the non-cooperative IO signal (e.g., communication signal, broadcasting signal, or navigation signal) for passive bistatic radar (PBR) applications has been thoroughly demonstrated in the last decades [22,23]. Different techniques have been developed for PBR, such as moving target range-Doppler mapping and tracking, inverse SAR imaging, ground/maritime moving target indication, and passive bistatic SAR imaging [24–32]. Compared to conventional active radar, several advantages can be achieved by PBR, such as no need for frequency allocation, smaller vulnerability, higher flexibility, and lower cost. Therefore, PBR has been viewed as an effective complementary technique to active radar and has been extensively developed for various applications, such as harbor protection, traffic density monitoring, through the wall moving target detection, and coherent change detection [33–36]. For PBR, the commonly used IOs can be summarized as [23] (a) terrestrial or satellite communication system, such as 4G long term evolution (LTE), Wireless Fidelity (WiFi), and Globalstar signal; (b) terrestrial or satellite broadcast system, such as FM radio, digital audio broadcast (DAB), and digital television signal; (c) global navigation signal system, such as GPS, GLONASS, Galileo, and BeiDou; and (d) other terrestrial or spaceborne radar systems used for remote sensing applications, such as TerraSAR-X.

Among these IO signals, a suitable one should be selected for GB-SAR applications. To satisfy the requirements of wide bandwidth, continuous illumination, and high-accuracy displacement estimation, continuous digital TV signal broadcast from a geostationary satellite is used in this study. Specifically, Japan communication satellite (CS) digital TV signal is used for PB-GB-SAR. Compared to other existing commercial signals (such as FM, DAB, and GNSS), satellite digital TV signal has a wider bandwidth, and multiple TV channels from different transponders can be combined to get a submeter range resolution. Compared to the PBR systems using the signal transmitted from other radar systems, its geostationary orbit provides the capability to continuously monitor a nationwide region. Besides, the working frequency of the satellite digital TV signal is typically from 10 to 13 GHz, providing a centimeter-level wavelength and a high-resolution displacement estimation capability. Furthermore, although different standards have been adopted (for example, ISDB-S and ISDB-S3 in Japan, DVB-S and DVB-S2 in Europe, and ABS-S in China), satellite digital TV signal can be received globally, giving the potential of using PB-GB-SAR at different geographical locations for different applications.

Although the basic principle of PB-GB-SAR seems simple and the properties, such as working frequency, resolution, and ambiguity function, of the geostationary satellite digital TV signal seem promising (as indicated in [37–41]), the real implementation of a PB-GB-SAR system for imaging and displacement estimation purposes has several problems. For example, for most PBR implementations, two channels (i.e., reference channel and surveillance channel) are inherently required to receive the illuminating signal and the target reflections. The coherence between the two channels is important to increase the signal to noise ratio (SNR) by using a long integration time or adding multiple short-term measurements together. Because commercial-off-the-shelf (COTS) hardware used for the reception

of satellite digital TV signal is easily available on the market, the first problem we faced is how to have two coherent (frequency and phase synchronized) channels based on COTS antennas, low noise blocks (LNBs), and amplifiers, without building a radar system from scratch to reduce the system cost. Furthermore, since PB-GB-SAR is a bistatic system, its geometry should be considered for the following imaging and interferometric processes. Although we have established a simplified geometry in the current study, PB-GB-SAR imaging of a local area still needs further studies. Another issue related to the bistatic geometry is the different concept of Line-of-Sight (LoS) displacement with respect to conventional monostatic GB-SAR systems. The geometry-related issues are thus the second problem we faced.

To solve these problems, we propose several methodologies. Firstly, we propose a synchronization method for multiple COTS LNBs with different local oscillators. The synchronization can help to increase the integration time of the reference and surveillance channels, which is important to get a sufficient SNR for target imaging. It has been observed that, under the proper conditions, the frequency/phase coherence is stable over a long period. Secondly, we modify the time-domain back-projection algorithm (BPA) and range-migration algorithm (RMA) using a simplified geometry and plane wave approximation to focus PB-GB-SAR images. Compared to BPA which is simple and easy to implement, RMA is more time-saving to image a local scene with a fine grid size and can obtain a comparable performance. At last, the formula of displacement estimation of the target imaged by PB-GB-SAR is derived. All the proposed methodologies are then validated by field experiments. The designed PB-GB-SAR system can provide a focused SAR image of different targets (such as fence, light pole, small house, and car) within a 100-meter range and the displacement of the target can be estimated with submillimeter accuracy.

The remainder of this paper is organized as follows. In Section 2, the designed PB-GB-SAR system, link budget analysis, and LNB synchronization method are presented. The signal model and two imaging methods for PB-GB-SAR are presented in Section 3. In Section 4, experiment results of imaging and displacement estimation are shown. In Section 5, results are discussed and perspectives for future work are given. Finally, Section 6 concludes this paper.

## 2. System Overview

### 2.1. PB-GB-SAR Concept

A conventional GB-SAR system is shown in Figure 1, where the transmitter and receiver are both moved along a rail to generate a synthetic aperture. As a dedicated transmitting unit is used, the system is expensive. The purpose of this study is to replace the transmitter with an existing non-cooperative illuminator (e.g., a TV satellite) to develop a PB-GB-SAR system with only receivers. PB-GB-SAR can not only reduce the system cost but also improve the system flexibility, as illustrated in Figure 2, based on which the three-dimensional (3D) displacement vector of the target (e.g., a dam) can be obtained by coherently processing all the received signals. Moreover, low-cost and low-weight receivers can be deployed on different platforms at separated locations to observe the spatial diversity of the target scattering properties.

In this study, we used the Communication Satellite (N-SAT-110) on the geostationary orbit of the 110 degrees east longitude (CS 110° E) for digital TV transmission (with the ISDB-S standard via right-hand and left-hand circular polarizations) as a non-cooperative illuminator. Limited by the hardware, only the right-hand circularly polarized transmitted signal is received. The frequency of the CS digital TV signal is within the Ku band, from approximately 12.27 to 12.75 GHz with 12 channels and a bandwidth of about 474.5 MHz (each channel has a bandwidth of 34.5 MHz), which can provide a maximum range resolution of 0.32 m. The spatial coverage of the CS digital TV signal is shown on the left of Figure 3 with different equivalent isotropically radiated powers (EIRPs) and the spectrum of the real-sampled CS digital TV signal is shown on the right of Figure 3, where 12 channels can be clearly observed.

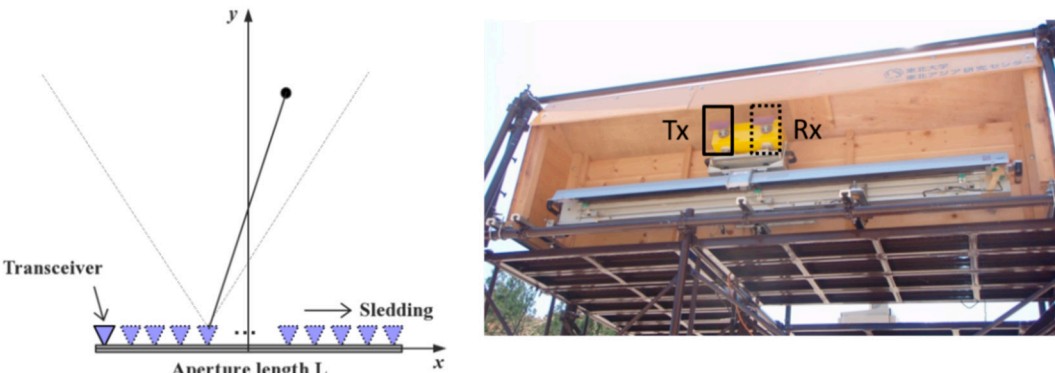

**Figure 1.** Conventional ground-based synthetic aperture radar (GB-SAR) system: (**left**) the imaging geometry and (**right**) a real stepped-frequency continuous waveform (SFCW)-based GB-SAR system used for landslide monitoring [4].

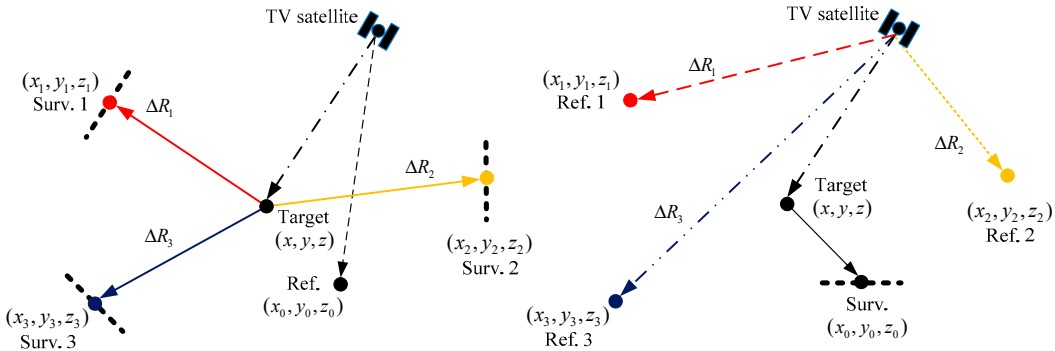

**Figure 2.** Passive bistatic (PB)-GB-SAR with multiple receivers for 3D displacement vector estimation. Note the difference between the left and right subfigures with different numbers of reference and surveillance antennas. More configurations can be used for specific applications.

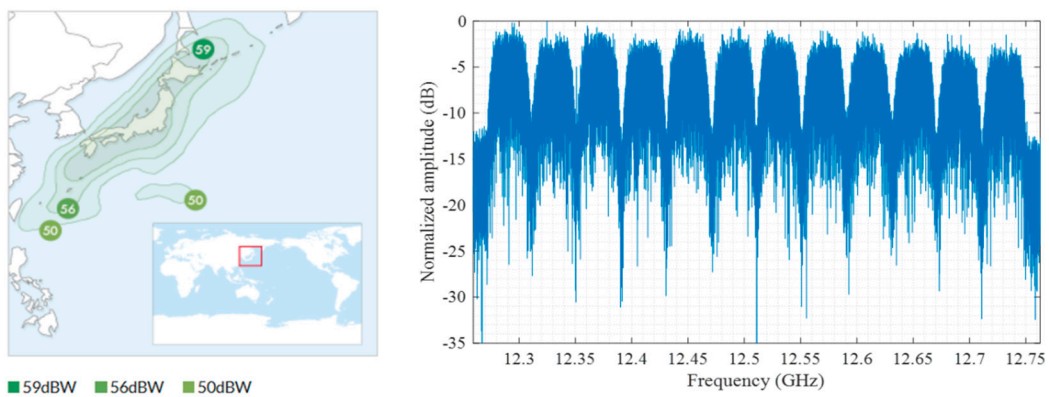

**Figure 3.** Communication satellite (N-SAT-110) digital TV signal: (**left**) spatial coverage with different EIRPs [42] and (**right**) spectrum of the real-sampled data.

The imaging geometry of the PB-GB-SAR system using satellite digital TV signal is shown in the left subfigure of Figure 4, where, similar to most PBR implementations, two channels (i.e., reference and surveillance channels) are used to receive the illuminating signal and the target reflections. The reference antenna is at a fixed position and its looking angle is fine-tuned to the satellite. In such a case, the $y$ direction can be defined by the line from the TV satellite to the reference antenna. In order to achieve a high azimuth resolution, the surveillance antenna is moved along a linear rail. Therefore, the $x$ direction (the wavefront of the satellite digital TV signal) can be defined as perpendicular to the $y$ direction on the plane determined by the antenna moving direction and the $y$ direction. In practical implementations, the reference antenna can be moved together with the surveillance antenna, which

can help to simplify the imaging process. The aim is to make the antenna moving direction parallel with the *x* axis to more easily determine the antenna position, which will be detailed in Section 3 and is shown in the right subfigure of Figure 4.

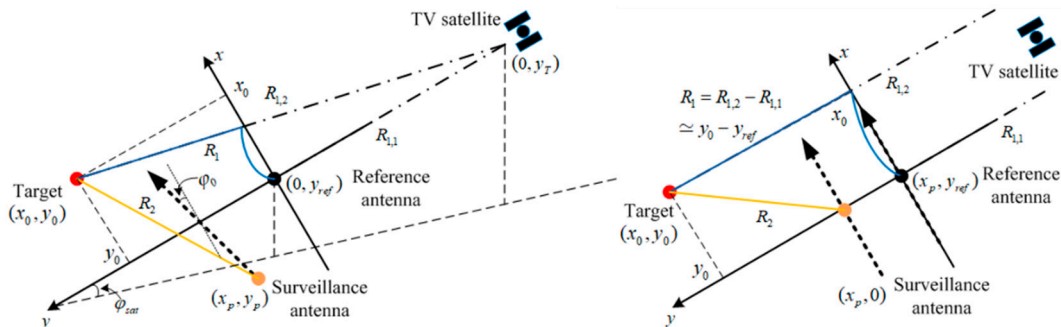

**Figure 4.** The imaging geometry of PB-GB-SAR: (**left**) the general geometry without the requirement of the antenna moving direction and (**right**) the simplified geometry with plane wave approximation.

To prove the concept with a low-cost set-up, we would like to use the COTS hardware that is easily obtainable on the market to build a PB-GB-SAR system. For the reception of satellite digital TV signal, several COTS components can be used, such as parabolic antenna, LNB, and amplifier (a.k.a. "booster" on the COTS market). Apart from these, we also need a data sampling component (a digital oscilloscope is used in this study), a positioner used to move the antenna to generate a synthetic aperture, and a controller to remotely control the data acquisition and antenna moving.

Based on the above-described hardware components, we have designed a PB-GB-SAR system, as shown in Figure 5, where the reference antenna is a 45 cm parabolic antenna with a gain of ~34 dB and a working frequency from 11.71 to 12.75 GHz. As to the surveillance channel, only the feed-horn of the parabolic antenna is used in order to achieve a wider beam-width for synthetic aperture processing. Two LNBs with different local oscillators are synchronized by an oscillator controller, which will be presented in detail in Section 2.3. Two boosters with a working frequency from 10 to 2600 MHz and a gain from 26 to 34 dB adjustable were used to amplify the received signal. The surveillance antenna is mounted on a programmable positioner to generate a synthetic aperture with a fixed step in the horizontal and/or vertical directions. The signal flow is sampled by a four-port digital storage oscilloscope (Agilent 54855A Infiniium Oscilloscope) with a sample rate of 10 GSamples/s: since the LNB will downconvert the frequency of the satellite digital TV signal to 1 GHz to 2 GHz, a 10 GSamples/s sampling rate is needed to avoid the distortion of the signal sampled by the oscilloscope. Finally, the dataset is transferred to a local PC for off-line signal processing, i.e., SAR imaging and displacement estimation.

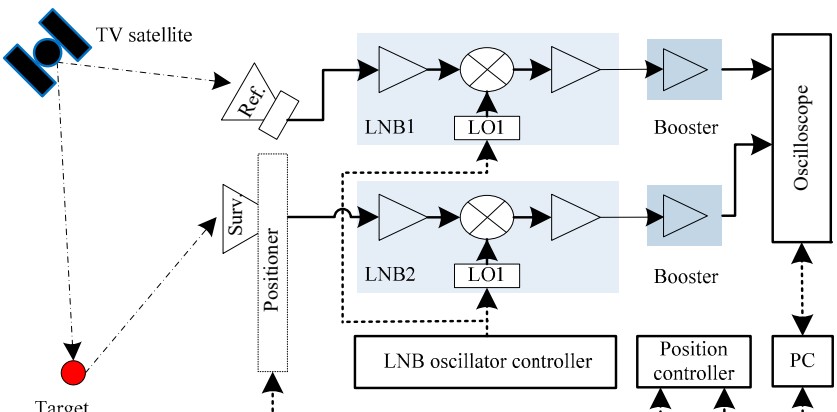

**Figure 5.** The designed PB-GB-SAR system, where the low noise block (LNB) oscillator controller is used to synchronize the phase and frequency of two LNBs, which will be detailed in Section 2.3.

## 2.2. Link Budget Analysis

Considering the power of the satellite digital TV signal on the Earth surface is low, this subsection analyzes the link budget of PB-GB-SAR to provide the necessary fundamentals for system parameter selection and signal processing.

As shown in Figure 4, the power budget of the reference channel is similar to a communication link budget. Assuming the one-way distance between the satellite and the reference antenna is $R_{1,1}$, the power of the received reference signal from a single TV channel can be expressed as [38]

$$P_{ref} = \frac{EIRP \cdot G_{ref} \lambda^2}{(4\pi)^2 R_{1,1}^2} \tag{1}$$

where the typical value of EIRP is from 50 to 59 dBW, as shown in Figure 3, and $G_{ref}$ denotes the gain of the reference antenna; $\lambda = c/f_c$ is the wavelength with $c$ as the speed of light and $f_c$ as the carrier frequency.

For the surveillance channel, the signal power is given by [39]

$$P_{surv} = \frac{EIRP \cdot G_{surv} \lambda^2 \sigma}{(4\pi)^3 R_{1,2}^2 R_2^2} \tag{2}$$

where $R_{1,2}$ is the distance between the TV satellite and the target, $R_2$ is the distance between the target and the surveillance antenna, $G_{surv}$ is the gain of the surveillance antenna (we note that, since only the feed-horn of the parabolic antenna without the reflector is used for the surveillance channel, as shown on the right of Figure 8, $G_{surv}$ is much smaller than $G_{ref}$), and $\sigma$ denotes the Radar Cross Section (RCS) of the target. Given the system noise temperature $T_0$ and the system noise bandwidth $B_0$, the noise power can be written as [39]

$$P_n = k_0 T_0 B_0 \tag{3}$$

where $k_0$ is the Boltzmann constant.

According to Equations (1)–(3), and using the parameters in Table 1, the SNR of the reference and surveillance channels can be obtained by

$$SNR_{ref} = P_{ref}/P_n = \frac{EIRP \cdot G_{ref} \lambda^2}{(4\pi)^2 R_{1,1}^2 k_0 T_0 B_0 L_r} = 10.1 \ dB \tag{4}$$

and

$$SNR_{surv} = P_{surv}/P_n = \frac{EIRP \cdot G_{surv} \lambda^2 \sigma}{(4\pi)^3 R_{1,2}^2 R_2^2 k_0 T_0 B_0 L_r} = -49.9 \ dB \tag{5}$$

It can be seen from Equations (4) and (5) that the SNR of the reference signal is more than 10 dB and can be used to form a near-optimal matched filter for range compression [43] (the small integration gain loss caused by the noise can be ignored), while the SNR of the surveillance signal is quite low and thus the TV spectrum cannot be explicitly observed. In Table 1, the antenna moving step is selected as 5 mm, which is smaller than a quarter of the wavelength. The synthetic aperture length and the coherent integration time are to be determined (TBD) by the following simulations.

Firstly, based on the matched filtering theory and by using multiple TV channels, the SNR after range compression obtained by cross-correlating the surveillance signal with the reference signal is given by [44]

$$SNR = \frac{EIRP \cdot G_{surv} \lambda^2 \sigma}{(4\pi)^3 R_{1,2}^2 R_2^2 k_0 T_0 L_r} N T_{int} \tag{6}$$

where $N$ is the number of used TV channels. With the parameters in Table 1, the SNR of the range compression result versus the integration time $T_{int}$ is shown on the left of Figure 6. Due to the memory

limitation of the used oscilloscope (only 1 Mpts deep memory is available for a data sampling rate of 10 GSamples/s); an integration time of 100 μs is used in this study. As we can see, to get an SNR higher than 0 dB after range compression, the minimal integration time should be ~236 μs, which cannot be obtained by the used oscilloscope in a single measurement (summing multiple measurements together is however possible). With a 100 μs coherent integration time, the SNR is still lower than 0 dB (−3.74 dB) and the target cannot be detected. Therefore, azimuth compression is further conducted. After both range and azimuth compression, the SNR of the SAR image is given by [44,45]

$$SNR = \frac{EIRP \cdot G_{surv}\lambda^2\sigma}{(4\pi)^3 R_{1,2}^2 R_2^2 k_0 T_0 L_r} NT_{int}P \tag{7}$$

where $P = [L/\Delta x + 1]$ denotes the number of azimuth samples. With a 100 μs coherent integration time, the SNR of the SAR image versus the synthetic aperture length is shown on the right of Figure 6, from which it can be learned that to get an SNR higher than 20 dB (a high SNR is critical to get a high precision in displacement estimation, and, in practical applications, the SNR will be reduced by many nonoptimal factors), the minimal synthetic aperture length should be ~1.18 m. In this study, a 1.2 m synthetic aperture length is selected to balance the achievable SNR level and the data acquisition time. Due to the antenna movement along the positioner and the data transmission from the oscilloscope to the PC, the data acquisition time for 241 measurements (1.2 m with 5 mm step) is ~15 min.

**Table 1.** Parameters used for link budget analysis.

| Parameter | Symbol | Value | Unit |
|---|---|---|---|
| TV signal power | EIRP | 55 | dBW |
| Reference antenna gain | $G_{ref}$ | 34 | dB |
| Speed of light | $c$ | $3 \times 10^8$ | m/s |
| Carrier frequency | $f_c$ | 12.51 | GHz |
| Wavelength | $\lambda$ | 24 | mm |
| Direct path | $R_{1,1}$ | 36,000 | km |
| Surveillance antenna gain | $G_{surv}$ | 15 | dB |
| Target RCS | $\sigma$ | 10 | m$^2$ |
| Satellite-target distance | $R_{1,2}$ | 36,000.1 | km |
| Target–receiver distance | $R_2$ | 100 | m |
| Noise temperature | $T_0$ | 290 | K |
| Noise bandwidth | $B_0$ | 34.5 | MHz |
| Boltzmann constant | $k_0$ | $1.38 \times 10^{-23}$ | J/K |
| Number of TV channels | $N$ | 12 | 1 |
| Path loss of Ku band | $Lr$ | 2 | dB |
| Synthetic aperture length | $L$ | **TBD** | m |
| Antenna moving step | $\Delta x$ | 5 | mm |
| Coherent integration time | $T_{int}$ | **TBD** | μs |

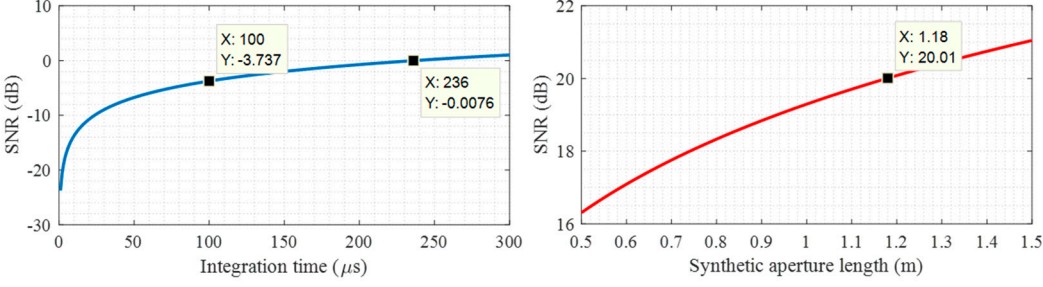

**Figure 6.** The SNRs of (**left**) the range compression result versus the integration time and (**right**) the SAR image versus the synthetic aperture length.

With a 100 µs coherent integration time and a 1.2 m synthetic aperture length, the target SNRs for different RCSs and target–receiver distances are shown in Figure 7. Because of the limitations introduced by the COTS hardware components, especially the low-gain surveillance antenna and the limited oscilloscope memory, we will focus on the targets within a 100 m range. This is a limitation of our current PB-GB-SAR system that would need an improvement in the future by using more dedicated hardware components.

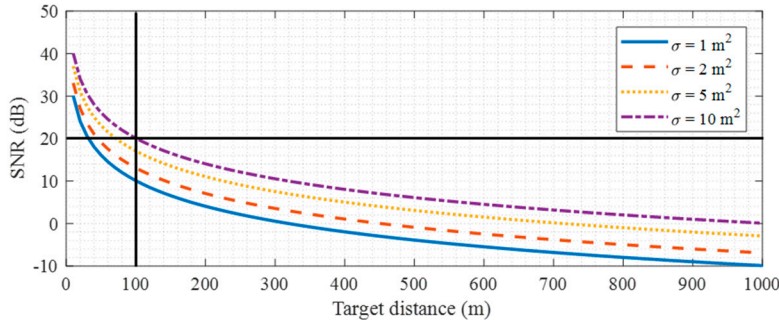

**Figure 7.** Target SNRs versus different RCSs and target–receiver distances. With the current PB-GB-SAR system, we have focused on the targets within a 100 m range.

### 2.3. LNB Synchronization

As indicated in the previous subsection, to get a sufficient SNR of PB-GB-SAR imaging, a long coherent integration time (100 µs) is needed. However, two COTS LNBs of reference and surveillance channels have different local oscillators, resulting in different frequencies and phases of the received signals. Then, if a longer integration time is used, the SNR will not be increased but instead be reduced because the two channels are no longer coherent. For a single measurement where the phase information is not important, this problem can be solved by the Fourier-transform based method to estimate the frequency difference between two LNBs [37]. However, for PB-GB-SAR imaging, in order to sum multiple short-term measurements together to improve the SNR for range compression and coherently process all the data from different antenna positions for azimuth compression, the phase difference between the two LNBs of different measurements should also be corrected. When a strong, stable, and stationary target with a known position can be found in the illuminated scene, the opportunity-target-based method proposed in [46] can be used for the phase correction. In spite of its low computational complexity and hardware cost, this method may hardly be used in the real conditions where strong and known targets are not available. Therefore, we decided to modify two COTS LNBs to make their frequency and phase synchronized.

Figure 8 shows the setup of the proposed LNB synchronization approach, where only one LNB is shown in the right subfigure and the diagram of the clock distribution circuit (the left subfigure) is shown in Figure 9. Based on the unpublished application note "X-tal driver for using multiple TFF10xx to the same X-tal as reference" describing the mechanism for locking multiple TFF10xx phase locked loops (PLLs) on the same reference as desired, a dedicated Pierce oscillator was built around a 25 MHz resonator and a 74HC04 inverter. When an external local oscillator is available, the Pierce oscillator can be replaced to provide a more stable and precise clock signal. The output of the Pierce oscillator (or the input local oscillator signal) feeds another 74HC04 used to generate three sets of in-phase and phase-opposition (180-degree phase shift) signals feeding the PLL inputs of the LNBs (three sets of signals can be used for synchronization of three LNBs, while, in this paper, only two sets of signals are used). Care must be taken on the one hand to tune the output voltage amplitude to acceptable levels (500 mVpp) using a voltage divider bridge and, on the other hand, to include a DC-blocking capacitor (5.6 nF) between the oscillator output and the PLL to avoid DC-current leakage from the PLL, which would prevent PLL locking.

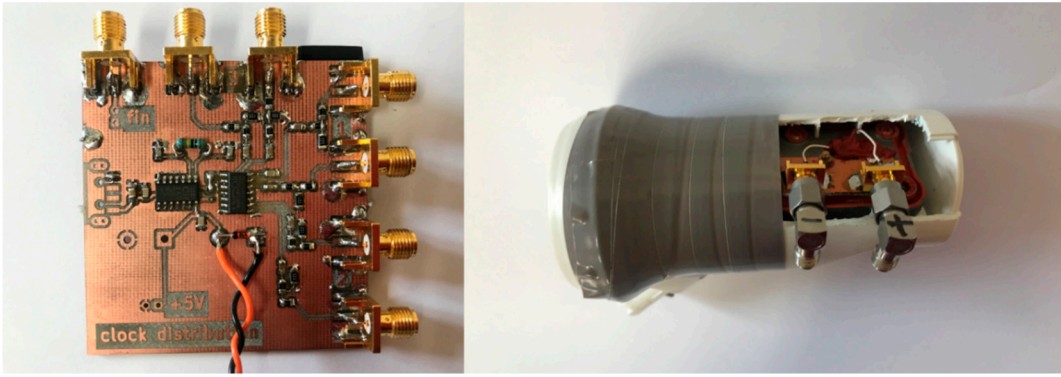

**Figure 8.** LNB synchronization setup: (**left**) clock distribution circuit with three sets of in-phase and phase-opposition signal outputs feeding the phase locked loop (PLL) inputs of LNBs and (**right**) the modified feed-horn of a 45-cm parabolic antenna receiving the in-phase and phase-opposition signals.

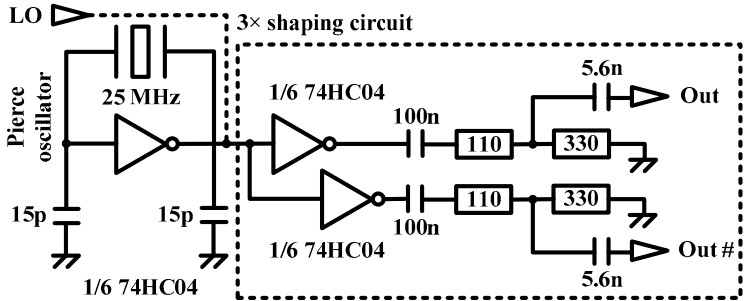

**Figure 9.** Diagram of the clock distribution circuit. Note that only one of "LO" or "Pierce oscillator" is needed. When an external LO is available, it can be used to provide more stable and precise results. In this study, the Pierce oscillator is used.

It has been observed by experiments that, under proper conditions, frequency/phase coherence between two LNBs could be stable over tens of minutes, as needed for PB-GB-SAR imaging and displacement estimation. For example, by facing two synchronized parabolic antennas to the TV satellite, as shown on the left of Figure 10, the coherence between two channels is measured by analyzing the amplitude and phase stability of the cross-correlation peak by using all the 12 TV channels. On the right of Figure 11, the time delay between two channels is estimated to be about −0.165 m. Four smaller peaks at about ±7.5 m and ±15 m are generated by the amplitude-filtered spectrum and the frequency gaps of the satellite digital TV signal, which can be suppressed by some advanced methods.

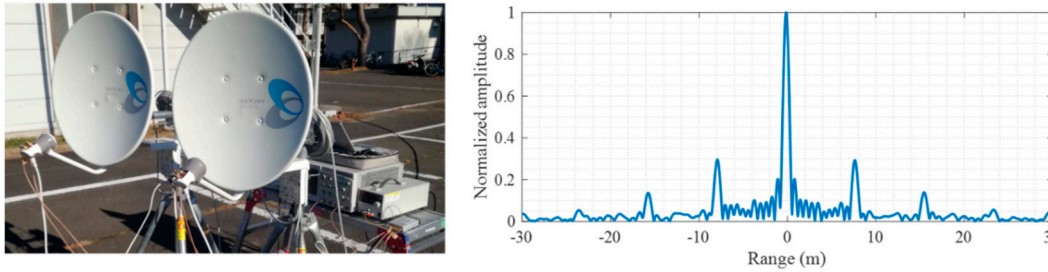

**Figure 10.** (**left**) Time delay measurement setup and (**right**) estimation result. The coherent integration time of each measurement is 10 μs, while, due to the delay caused by the data transmission from the oscilloscope to the PC, the measurement period is ~0.18 s. Therefore, for 5000 measurements, ~15 min is needed.

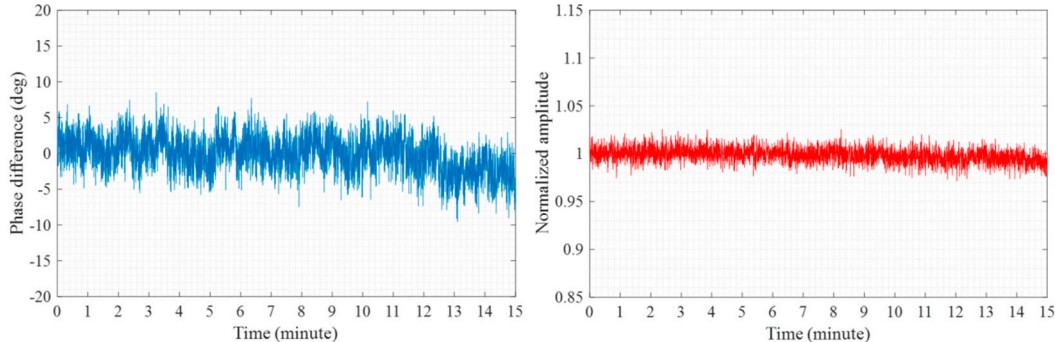

**Figure 11.** (**Left**) phase and (**right**) amplitude difference of the cross-correlation peaks for 15 min.

The phase and amplitude stability measurement results are shown in Figure 11. It can be learned that the phase difference (the first measurement is acted as the reference) over 15 min is within the interval of [−10, +10] degrees, the mean value is approximately −0.05 degrees and the standard deviation is ~2.52 degrees. Considering a wavelength of 24 mm, these two phases correspond to displacements of −1.67 μm and +84 μm, which can be ignored for displacement estimation with submillimeter accuracy. The normalized amplitude (similarly, the first measurement is acted as the reference) has a mean value of ~0.9986 and a standard deviation of ~0.0077. These results indicate that the proposed method can help to synchronize the frequency and phase of different LNBs, providing two coherent channels for PB-GB-SAR imaging. In addition, the phase stability also shows that the phase difference caused by the COTS boosters can be assumed to be constant. The amplitude stability also indicates that the transmitting power of the TV signal is stable over the measurement time.

## 3. Signal Processing

In this section, the signal model of PB-GB-SAR is established based on the geometries in Figure 4. Then, BPA- and RMA-based methods are modified for PB-GB-SAR imaging. At last, the displacement estimation formula is derived.

### 3.1. Signal Model

For the $n$-th channel with $n$ = 1, 2, ..., $N$ ($N$ = 12), the satellite digital TV signal can be expressed as

$$s_0^n(t) = \sum_{i=-\infty}^{\infty} A_n \exp(j\varphi_i^n) h(t - iT_s) \exp(j2\pi f_c^n t) \tag{8}$$

where $t$ is time; $A_n$ is the amplitude; $h(t)$ is the impulse response of the root-raised-cosine (RRC) filter [47] with a duration of $T_s$; $f_c^n$ is the carrier frequency; and $\varphi_i^n$ is the phase of the $i$-th symbol. For QPSK modulation, $\varphi_i^n$ equals to $\pi/4$, $3\pi/4$, $5\pi/4$, or $7\pi/4$.

The frequency transfer function of the RRC filter can be expressed as [47]

$$H(f) = \begin{cases} 1 & |f| \le f_s(1-\alpha) \\ \sqrt{\frac{1}{2} + \frac{1}{2}\sin\frac{\pi}{2f_s}\left(\frac{f_s-|f|}{\alpha}\right)} & f_s(1-\alpha) \le |f| \le f_s(1+\alpha) \\ 0 & |f| \ge f_s(1+\alpha) \end{cases} \tag{9}$$

where $fs$ = 0.5/$Ts$ is the Nyquist frequency and $\alpha$ = 0.35 is the roll-off factor.

Based on Equations (8) and (9), the spectrum of the *n*-th TV channel with an integration time of $T_{int}$ can be expressed as

$$s_0^n(f) = \left( H(f - f_c^n) \sum_{i=-\infty}^{\infty} A_n e^{j\varphi_i^n} e^{-j2\pi(f-f_c^n)iT_s} \right) * \left( T_{int} e^{-j\pi T_{int}(f-f_c^n)} \sin c[T_{int}(f - f_c^n)] \right) \tag{10}$$

where * denotes the convolution process. It can be learned from Equation (10) that the spectrum of each TV channel is determined by the spectrum of RRC filter and a *sinc* function. When a long integration time is used, the main lobe of the *sinc* function is narrow and the spectrum of each TV channel is dominated by the spectrum of RRC filter. In other words, the spectrum is amplitude-filtered by Equation (9) in such a case.

According to Figure 4, and with the amplitudes $A_{ref}$ and $A_{surv}$ that are independent of the frequency, the received reference signal and surveillance signal from all the TV channels can be expressed as

$$s_{ref}(t) = \sum_{n=1}^{N} A_{ref} s_0^n[t - R_{1,1}/c] \tag{11}$$

and

$$s_{surv}(t) = \sum_{n=1}^{N} A_{surv} s_0^n[t - (R_{1,2} + R_2)/c] \tag{12}$$

Range compression can be conducted by the cross-correlation technique, which will be more easily computed in the frequency domain by fast Fourier-transform (FFT). The frequency-domain reference signal and surveillance signal are given by

$$s_{ref}(f) = \sum_{n=1}^{N} A_{ref} s_0^n(f) \exp(-j2\pi f R_{1,1}/c) = A_{ref} s_0(f) \exp(-jkR_{1,1}) \tag{13}$$

and

$$s_{surv}(f) = \sum_{n=1}^{N} A_{surv} s_0^n(f) \exp[-j2\pi f(R_{1,2} + R_2)/c] = A_{surv} s_0(f) \exp[-jk(R_{1,2} + R_2)] \tag{14}$$

where $k = 2\pi f/c$ denotes the wavenumber and $s_0(f)$ is the spectrum of all TV channels.

Then, the bistatic time delay of the target can be calculated by the inverse Fourier-transform of the following frequency domain signal.

$$s(f) = s_{surv}(f)s_{ref}^*(f) = A_0 |s_0(f)|_2 \exp[-jk(R_1 + R_2)] \tag{15}$$

where $(\cdot)^*$ denotes complex conjugate, $R_1 = R_{1,2} - R_{1,1}$, and $A_0 = A_{surv} A_{ref}^*$.

For PBR, the inverse filtering (i.e., reciprocal filtering)-based method [29,32] can be used to avoid the influence of the IO waveform, i.e., the influence of $s_0(f)$ in Equation (15), resulting in

$$s_{inv}(f) = \frac{s_{surv}(f)}{s_{ref}(f)} = \frac{A_{surv}}{A_{ref}} \exp[-jk(R_1 + R_2)] \tag{16}$$

Since the inverse filtering needs a pure reference signal and its mismatched filtering property will reduce the output SNR, Equation (16) can only be used in the conditions where a template reference signal can be obtained and the SNRs of targets are high enough. Alternatively, the signal components with frequencies satisfying $|f| \leq f_s(1 - \alpha)$ can be extracted from each TV channel, i.e., only the flat spectrum can be used, to reduce the waveform influence of the used satellite digital TV signal (the amplitude-filtered spectrum), as indicated by Equation (9). However, this will also decrease the output SNR as the integration gain generated by the matched filtering is reduced. Therefore, in the

concept-proof study stage, Equation (15) is used for the following process without considering the spectrum properties of the satellite digital TV signal.

## 3.2. Imaging and Displacement Estimation

As mentioned previously, since a high azimuth resolution is desired, the surveillance antenna is moved along a positioner to generate a synthetic aperture, while the reference antenna is stationary and its looking angle is adjusted to the satellite, as shown on the left of Figure 4. For the *p*-th position of the surveillance antenna $(x_p, y_p)$, the range compression result obtained by combing *N* TV channels is given by

$$\chi_p = F^H s_p(f) = F^H [s_{surv}^p(f) s_{ref}^p(f)^*] \tag{17}$$

where *F* is the Fourier-transform matrix.

Since the targets within short ranges (i.e., 100 m) are considered, it is not necessary to calculate the time delays corresponding to the far range targets. Therefore, Equation (17) is modified to

$$\chi_p = F^H s_p(f) = F^H \sum_{m=1}^{M} s_p^m(f) = F^H \sum_{m=1}^{M} s_{surv}^{p,m}(f) s_{ref}^{p,m}(f)^* \tag{18}$$

where $M = T_{int}/T_p$, with $T_p$ being the processing time. By doing so, the computational complexity can be reduced for the following imaging methods.

After range compression at each antenna position, azimuth compression can be conducted by the time domain BPA, which estimates the complex amplitude of the target at $(x_0, y_0)$ as

$$\sigma_{BPA}(x_0, y_0) = \sum_{p=1}^{P} \chi_p [R_1(x_0, y_0) + R_2^p(x_0, y_0)] \tag{19}$$

where

$$R_1(x_0, y_0) = R_{1,2}(x_0, y_0) - R_{1,1} = \sqrt{x_0^2 + (y_0 - y_T)^2} - (y_{ref} - y_T) \tag{20}$$

denotes the difference between the satellite-to-target distance and the satellite-to-reference-antenna distance, and

$$R_2^p(x_0, y_0) = \sqrt{(x_0 - x_p)^2 + (y_0 - y_p)^2} \tag{21}$$

denotes the distance between the target and the surveillance antenna.

It should be noted that, to calculate Equation (20), the position of the TV satellite should be known in advance, and, to calculate Equation (21), the surveillance antenna position should be determined by its moving step $\Delta x$ and the angle between its moving direction and the *x* axis, i.e., $\varphi_0$ on the left of Figure 4.

To simplify the calculation of Equations (20) and (21), it is desired that the angle $\varphi_0$ is zero, i.e., the antenna moving direction should be parallel with the *x* axis. To this end, the reference antenna faced to the satellite is moved together with the surveillance antenna, as shown on the right of Figure 4.

The peak amplitude stability of the autocorrelation function of the received reference signal can be checked to make sure that the antenna moving direction is always parallel with the *x* axis (the wavefront of the satellite digital TV signal). For example, Figure 12 shows that when the antenna moving direction is parallel with the wavefront, the normalized autocorrelation peak amplitude (the first measurement is acted as the reference) is stable along the synthetic aperture with a mean value of 0.9972 and a standard deviation of 0.0078. When the moving direction is inclined, the normalized amplitude will be changed along the synthetic aperture, so requiring for a correction.

According to the geometry shown on the right of Figure 4, Equation (21) can be modified to

$$R_2^p(x_0, y_0) = \sqrt{(x_0 - x_p)^2 + y_0^2} \tag{22}$$

where $x_p = -L/2 + (p - 1)\Delta x$ is the *p*-th antenna position, which is only dependent on the moving step.

Moreover, for the reference antenna with $(x_p, y_{ref})$ denoting its $p$-th antenna position, based on the plane wave approximation and Taylor series expansion, it can be derived that

$$R_1^p(x_0, y_0) = R_{1,2}(x_0, y_0) - R_{1,1}^p \simeq y_0 - y_{ref} \tag{23}$$

The maximal approximation error in Equation (23) will be increased for the target with a larger azimuth position $x_0$, as shown in Figure 13, where the simulation parameters are: $y_T = -36{,}000$ km and $x_p$ is from −0.6 m to 0.6 m with a step of 5 mm. It can be seen that, with the considered local area imaging, the approximation error is always smaller than 0.15 mm (corresponding to a phase error of $\pi/80$). Therefore, the plane wave approximation is used for PB-GB-SAR imaging to simplify the imaging process, as, in such a case, the TV satellite position is not necessary to be known in advance.

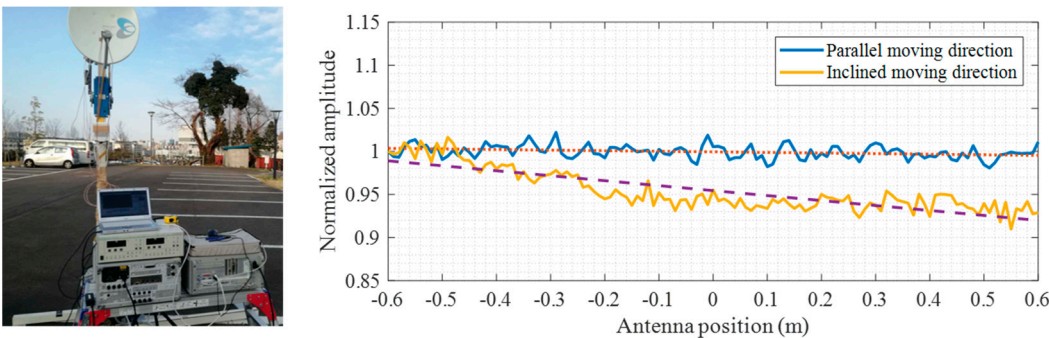

**Figure 12.** Using the stability of the peak amplitudes of reference signal autocorrelation function to determine the antenna moving direction: (**left**) experiment setup and (**right**) calculation results.

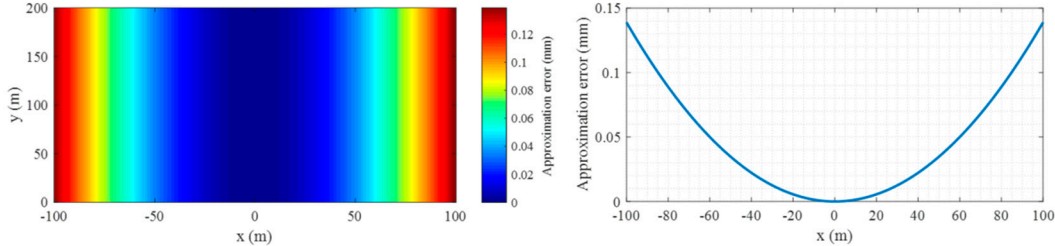

**Figure 13.** Approximation error for targets at different positions. The right subfigure is for the targets with $y_0 = 100$.

Then, the complex amplitude of the point at $(x_0, y_0)$ can be estimated by BPA as

$$\sigma_{BPA}(x_0, y_0) = \sum_{p=1}^{P} \chi_p [R_1^p(x_0, y_0) + R_2^p(x_0, y_0)] \tag{24}$$

To effectively calculate Equation (24), the FFT of zero-padded $s_p(f)$ with the interpolation process or the type-II nonuniform FFT (NUFFT) [48] can be used. In this paper, the latter method is adopted, which can be expressed as

$$\sigma_{BPA}(x_0, y_0) = \sum_{p=1}^{P} \left[ NUFFT_{II}^{x_0, y_0} \left( s_p(f) \right) \right] \tag{25}$$

However, when a fine grid size is selected for imaging, the computational complexity of Equation (25) is high. In the following, similar to GB-SAR, RMA is formulated for PB-GB-SAR.

At first, the frequency domain received signal from the whole observation scene at the $p$-th antenna position is expressed as

$$S(x_p, k) = \iint_{\Omega} \sigma(x, y) \exp\left\{ -jk(y - y_{ref} + R_2^p) \right\} dx dy \tag{26}$$

where σ(*x*, *y*) is the reflection coefficient of the target at (*x*, *y*) and Ω denotes the observation scene. Then, by Fourier-transform, with respect to $x_p$, we can get

$$S(k_x, k) = \int S(x_p, k) \exp\{-jk_x x_p\} dx_p = \iint_\Omega \sigma(x, y) S_1(k_x, k) dx dy \qquad (27)$$

where

$$S_1(k_x, k) = \int \exp\{-j[k(y - y_{ref} + R_2^p) + k_x x_p]\} dx_p \qquad (28)$$

Based on the principle of stationary phase (PSP), $S_1(k_x, k)$ can be approximated by

$$S_1(k_x, k) = \exp\left\{-j(\sqrt{k^2 - k_x^2} + k)y\right\} \exp\{-jk_x x\} \exp\{jky_{ref}\} \qquad (29)$$

By substituting Equation (29) into Equation (27), we have

$$S(k_x, k) = \exp\{jky_{ref}\} \iint_\Omega \sigma(x, y) \exp\{-jk_x x\} \exp\left\{-j(\sqrt{k^2 - k_x^2} + k)y\right\} dx dy \qquad (30)$$

Conventionally, to use FFT to increase the computing speed, the interpolation process is used to form a signal matrix uniformly sampled in the *kx-ky* domain, which can be expressed as

$$S(k_x, k_y) = S\left(k_x, k = (k_x^2 + k_y^2)/(2k_y)\right) \Leftarrow S(k_x, k) exp(-jky_{ref}) \qquad (31)$$

where the arrow stands for interpolation. It should be noted that, different from GB-SAR with $k_y = \sqrt{4k^2 - k_x^2}$, for PB-GB-SAR, we have $k_y = \sqrt{k^2 - k_x^2} + k$. Finally, according to Equation (31), SAR image can be obtained by the 2D inverse Fourier-transform as

$$\sigma_{RMA}(x, y) = F_x^H \big[ S(k_x, k_y) \big] F_y^* \qquad (32)$$

where *Fx* and *Fy* are the Fourier-transform matrices in the *x* and *y* directions, respectively.

Alternatively, instead of interpolation and then FFT in the *y* direction, the type-I NUFFT [48] can also be used, based on which Equation (32) can be modified to

$$\sigma_{RMA}(x, y) = F_x^H \big[ NUFFT_I^k \big( S(k_x, k) exp(-jky_{ref}) \big) \big] \qquad (33)$$

Compared to BPA, RMA can get a comparable result, while the computing time can be reduced for imaging a local area with fine grid size. However, it should be noted that for a large scene and a proper grid size, the computational complexity of BPA can be lower than RMA, since, in this case, zero padding is necessary along the azimuth direction for RMA, resulting in a lot of samples in the azimuth direction and a time-consuming 2D interpolation or 1D NUFFT process.

Based on BPA or RMA, a time series of SAR images can be obtained with a temporal baseline of about 15 min by the PB-GB-SAR system. For each pair of SAR images, an interferometric phase image can be generated by

$$\Delta\varphi_{D-In}(x, y) = \arg\{\sigma_{mater}^*(x, y) \cdot \sigma_{slave}(x, y)\} \qquad (34)$$

Then, the two-way displacement of the target can be estimated by $D = \lambda\Delta\varphi_{D-In}(x, y)/2\pi$. For GB-SAR, the relationship $D = 2 (\delta x \sin\theta + \delta y \cos\theta) = 2\delta r$ can be used, where δx and δy are the target displacements in the *x* and *y* directions and $\theta = \arctan(x/y)$ is the angle of the target. Therefore, for GB-SAR, the one-way target displacement along the LoS direction can be estimated as $\delta r = D/2$. For PB-GB-SAR, there is not a single LoS direction since the bistatic geometry is used. However, for the currently used geometry, based on the plane wave approximation and given that the target only has

the displacement along the LoS direction of the surveillance antenna (i.e., $\delta x / \delta y = \tan\theta$), the two-way displacement estimated by PB-GB-SAR can be expressed as

$$D = \delta y + \delta r = \delta r(1 + \cos\theta) \tag{35}$$

Therefore, the LoS displacement with respect to the surveillance antenna can be calculated by $\delta r = D/(1 + \cos\theta)$.

## 4. Experiment Results

In this section, experiment results are presented to assess the performance of the developed PB-GB-SAR system and proposed methods.

At the beginning, a simple target, i.e., a metallic plate at near range, as shown in Figure 14, is imaged by PB-GB-SAR. The synthetic aperture length is 1.2 m with a step of 5 mm, the integration time is 100 μs, and the processing time is 1 μs. Based on Equations (25) and (32), SAR images of the metallic plate obtained by BPA and RMA are shown in Figure 15. It can be seen that the target can be well focused, while strong artifacts are generated in the range direction. Although these two methods can get similar results, the computational complexity of RMA is smaller than BPA (the desired computing accuracy of type-I and type-II NUFFTs is set to be $10^{-12}$). RMA requires only ~0.79 s (measured by the TIC and TOC instruction in MATLAB on a Core i5, 2.5 GHz, and 8 GB RAM PC), while, although the grid size has been reduced 8-fold (2 times in the $x$ direction and 4 times in the $y$ direction), BPA still needs ~8.54 s to get the SAR image, indicating the efficiency of RMA in such a case.

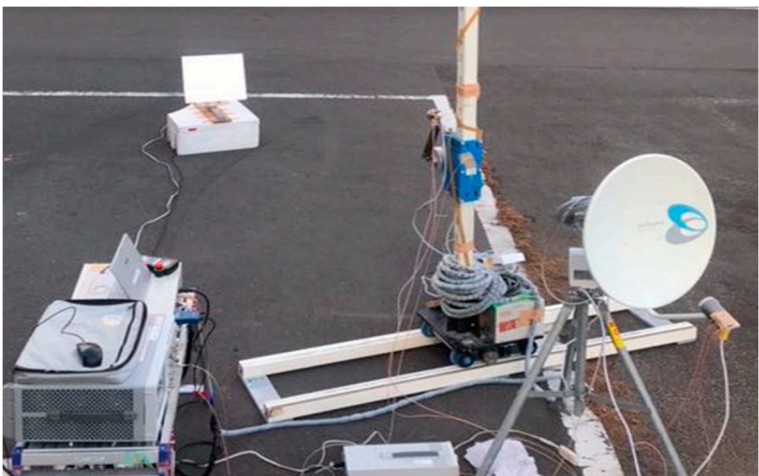

**Figure 14.** PB-GB-SAR imaging of a metallic plate: The plate angle is tuned to make its reflection maximal and the antenna moving direction is adjusted to be parallel with the wavefront of the TV signal.

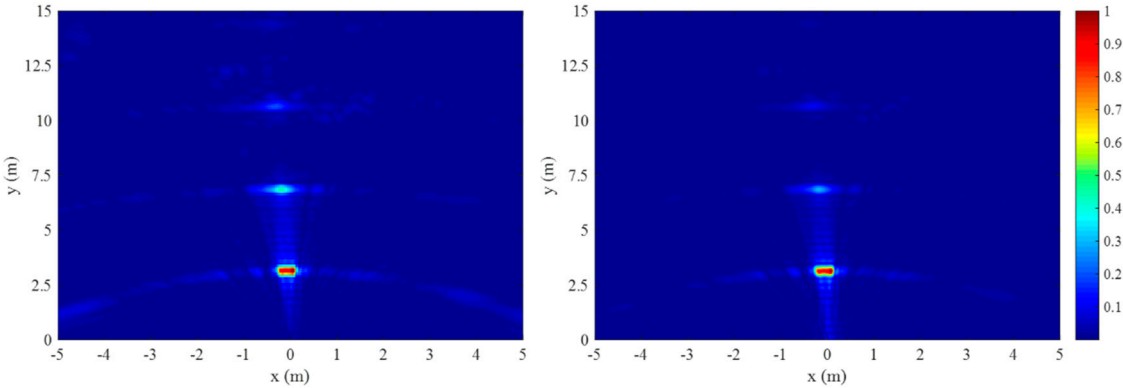

**Figure 15.** Normalized focused SAR images of the metallic plate obtained by (**left**) BPA and (**right**) RMA.

To assess the displacement estimation capability of the PB-GB-SAR system, the metallic plate is moved along a positioner from 1 mm to 15 mm with a step of 1 mm. The accumulated displacements estimated using the BPA and RMA-focused SAR images are shown in Figure 16. Since the angle of the metallic plate is close to zero, i.e., $\theta \approx 0$, as shown by the imaging result, its displacement along the LoS direction is about $\delta r = \lambda \Delta\varphi_{D\text{-}In}(x,y)/4\pi$, as given in Equation (35). It can be observed that both methods can accurately estimate the displacement: the root mean square errors (RMSEs) are 0.264 mm and 0.268 mm for BPA and RMA, respectively.

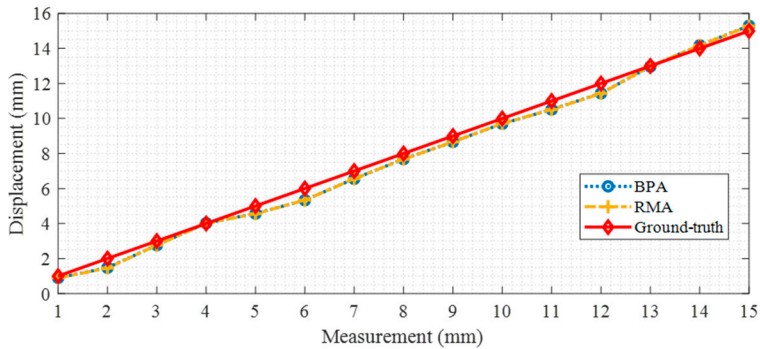

**Figure 16.** Displacement estimation of the metallic plate sledded on the rail.

To show the performance of the PB-GB-SAR system for imaging natural and man-made targets at short and middle range, a further experiment was conducted, as shown in Figure 17. Using the same 100 μs integration time, the processing time is increased to 4 μs due to the larger detection distance. To focus SAR images, BPA is used instead of RMA, as a lot of zero-padded samples in the azimuth direction make RMA significantly time-consuming, while BPA can use a larger grid size to reduce the computational complexity. The resulting image is shown in Figure 18, where the fence, light pole, tree, and small house can be imaged, as clearly shown in the right subfigure. Furthermore, the image in the left subfigure shows two cars located at approximately 60 m and 65 m in the parking lot, as indicated by the rectangle. By overlaying the SAR image with the aerial picture provided by Google maps, it can be seen that the obtained SAR image matches well with the real scene, as shown in Figure 19.

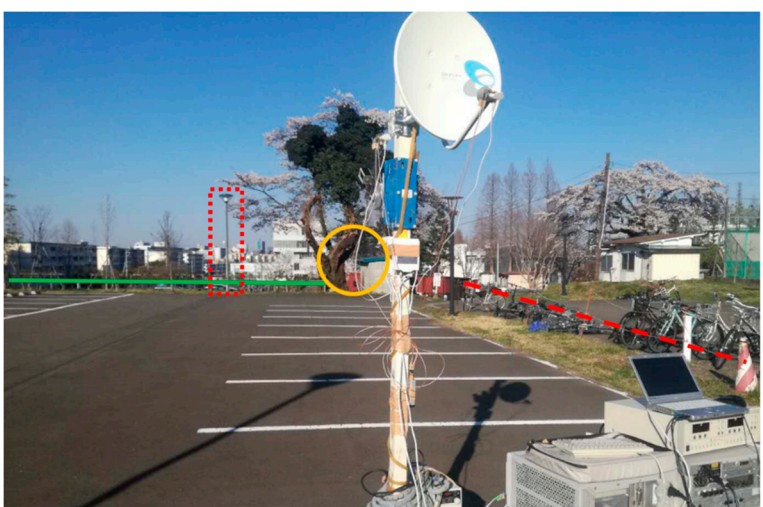

**Figure 17.** PB-GB-SAR imaging of natural and man-made targets: The green solid line indicates a fence, the dotted rectangle indicates a light pole, the yellow ellipse includes a tree and a small house, and the red dotted line indicates some bicycles.

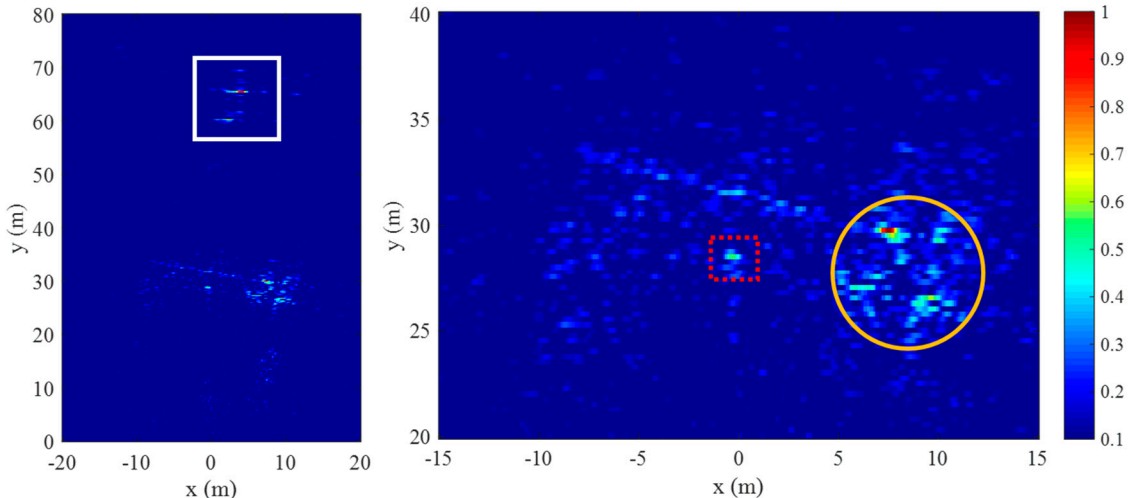

**Figure 18.** SAR image of natural and man-made targets: The right subfigure is the zoomed version of the left subfigure with specific focuses on the fence, light pole, tree, and house.

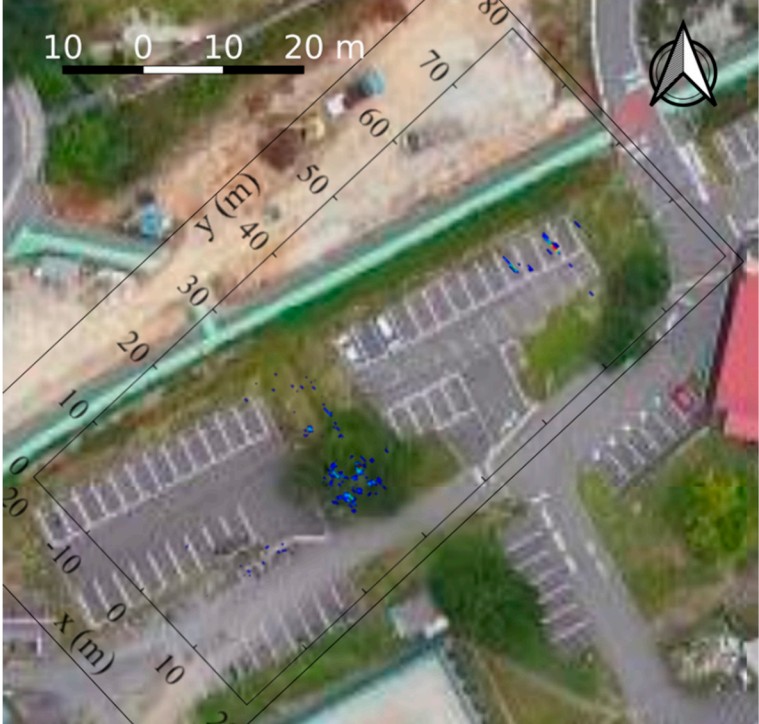

**Figure 19.** SAR image overlaid with the aerial view provided by Google maps. Note that the two strong targets at about 60 m and 65 m are located at a parking lot.

## 5. Discussions

### 5.1. Artifact Suppression

As shown in Figures 10 and 15, strong artifacts exist in the range direction, which are generated by the waveform structure of the used satellite digital TV signal. More specifically, the amplitude-filtered spectrum and the frequency gap between two adjacent TV channels, as shown in Figure 3, make the range compression nonoptimal. To reduce the influence of the amplitude-filtered spectrum,

as mentioned previously, the flat spectrum can be extracted for range compression. For each TV channel, according to Equation (9), the extraction operation can be expressed as

$$s_W(f) = W(f - f_c)s(f) \simeq W(f - f_c)A_0 \exp[-j2\pi f(R_1 + R_2)] \tag{36}$$

where $W(f)$ is a window function in the frequency domain, given by

$$W(f) = \begin{cases} 1 & |f| \le f_s(1 - \alpha) \\ 0 & otherwise \end{cases} \tag{37}$$

With respect to the frequency gaps, based on the known spectrum in Equation (36) and some assumed signal models, the gapped signal components can be estimated, i.e., the frequency gaps can be filled. For instance, spatially variant apodization (SVA) based [30], compressive sensing based [49], and low-rank matrix completion based [37] methods have been used for frequency gap filling in different PBR applications. In the following, the Super-SVA-based method [37] is introduced.

Super-SVA [50] is based on SVA that is an adaptive amplitude windowing method with the capability to effectively suppress the sidelobes of Fourier-transform without broadening the mainlobe [51]. According to this property, Super-SVA extrapolates the spectrum of a signal by iteratively using SVA and the inverse weighting operations. In our case, at each antenna position, the Super-SVA-based method can be applied to each TV channel to expand its bandwidth. Mathematically, at each antenna position, for the $n$-th ($n$ = 1, 2, ..., 12) TV channel, according to Equation (36), the windowed frequency domain signal can be expressed as

$$s_W^n(f) = W(f - f_c^n)A_0 \exp[-j2\pi f(R_1 + R_2)] \tag{38}$$

With the SVA weighting function $\boldsymbol{W}_{SVA}(f)$ in the frequency domain [37], the Fourier-transform of the $n$-th TV channel is given by

$$\boldsymbol{\chi}_{SVA}^n = \boldsymbol{F}_n^H[s_W^n(f) \odot \boldsymbol{W}_{SVA}(f)] \tag{39}$$

where $\boldsymbol{F}_n$ is the Fourier-transform matrix for the $n$-th TV channel. Then, by performing the inverse Fourier-transform, a band-unlimited signal can be obtained as

$$s_{SVA}^n(f) = \boldsymbol{F}_n \boldsymbol{\chi}_{SVA}^n \tag{40}$$

Defining an inverse weighting function in the frequency domain as $\boldsymbol{W}_{inv}(f)$, which is the inverse Fourier-transform of the mainlobe of a *sinc* function in the time domain, the spectrum extrapolated signal can be obtained as [37]

$$s_{Super-SVA}^n(f) = s_{SVA}^n(f)/W_{inv}(f) \tag{41}$$

which has a wider bandwidth than the original signal in Equation (38). However, its bandwidth cannot be infinite because the inverse process should be truncated to avoid the problem of singularities. After inverse weighting and truncating, the center portion of the extrapolated signal in Equation (41) should be replaced with the original signal in Equation (38) to improve the accuracy, giving the first-time gap filling result. This procedure should be repeated for several times to fill the entire frequency gap between two TV channels. Since two adjacent TV channels have the same gapped frequencies, a simple average process can be conducted to make the reconstruction more accurate.

By filling the frequency gaps via the Super-SVA-based method at each antenna position, SAR imaging based on BPA and RMA can be modified correspondingly and the artifacts can be effectively suppressed. For example, the SAR image of the metallic plate obtained by RMA after frequency gap filling is shown in Figure 20. It can be learned that, after using the Super-SVA-based method, the high-level artifacts can be suppressed, giving a better imaging quality, which is more clearly shown in the right subfigure of Figure 20.

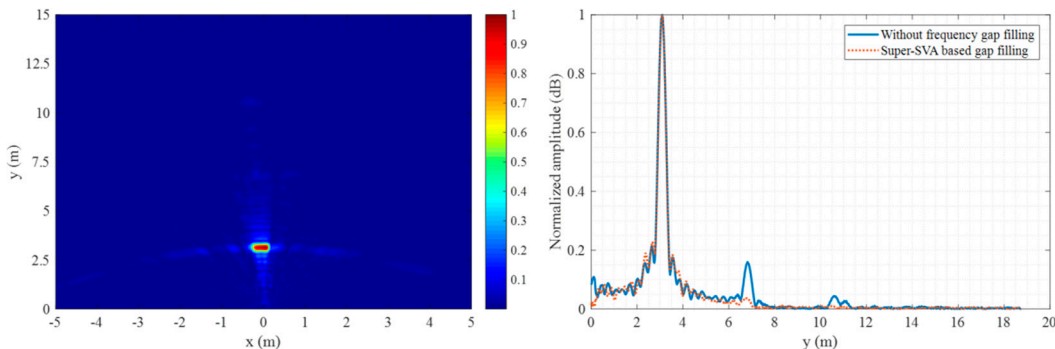

**Figure 20.** (**Left**) SAR image of the metallic plate obtained by RMA after frequency gap filling and (**right**) comparison between the results with and without gaps filling in the *y* direction.

Since each TV channel is filled separately at each antenna position, the relationship of different TV channels at different antenna positions is not exploited by the Super-SVA-based method. If all the frequency gaps for all the antenna positions can be processed at the same time, the gapped signal estimation accuracy can be increased [37]. Besides, when the SNR is low, the performance of the Super-SVA-based method with only the flat spectrum will be degraded. Reference signal reconstruction can be conducted to provide a template for inverse filtering to solve this problem. Therefore, as a key technique for the performance improvement of PB-GB-SAR, more advanced frequency gap filling methods should be applied.

Moreover, apart from the artifacts in the range direction, caused by the limited synthetic aperture length and frequency bandwidth, sidelobes are generated in the range and azimuth directions by the conventional matched filtering based imaging algorithms (BPA and RMA). To further improve the imaging quality with reduced sidelobe level, other advanced methods can be used, such as the sparse representation or compressive sensing-based [52,53] imaging method or the coherence factor or phase coherence factor-based [54,55] filtering method.

*5.2. Improvement Directions*

Although the performance of BPA and RMA based PB-GB-SAR imaging has been validated by experiments, they are not suitable to image a large observation area with fine grid size because of the high computational cost. Since the focus of this paper is on short-range and middle range targets, the computing burden of BPA or RMA is acceptable and thus other fast algorithms are not introduced. If a large scene is to be measured, some approximations can be used to derive faster imaging methods. For example, similar to the methods in [13,56], based on the far-field approximation and the condition that the synthetic aperture length is comparable to the range resolution, 2D FFT can be used to get the focused SAR image of the target.

Besides, some limitations of the developed PB-GB-SAR system are low SNR (and thus small coverage), narrow observation angle, long data acquisition period, and simple imaging geometry. The current implementation of PB-GB-SAR only uses COTS hardware to prove the concept. Dedicated hardware components, such as surveillance antenna with a wider beamwidth and high-quality low noise amplifiers should be used to improve the SNR and broaden the imaging angle. Since a digital oscilloscope is used to sample the data, which has a limited memory depth, an integration time longer than 100 μs is impossible by a single measurement. Besides, because of the data transmission from the oscilloscope to the PC, the data acquisition time should be further reduced. Therefore, more proper integrated digital data sampling platform should be used to increase the integration gain of matched filtering and increase the data acquisition rate. As a simplified imaging geometry is adopted in the current experiment, the potentials of PB-GB-SAR have not been fully exploited. Although it is a bistatic system, the obtained scattering properties of the imaging scene by the current PB-GB-SAR system is similar to the case of the conventional monostatic GB-SAR system. Therefore, more advanced

geometries should be adopted to get different bistatic and monostatic SAR images of the observation scene. Moreover, the system should not be limited on the ground; the low-cost and light receivers can be mounted on different platforms, such as an airship [57], to measure different displacement components of the targets.

Actually, the LNB synchronization circuit can provide three coherent channels for measurement, as shown in Figure 8, while only two of them are used in the current setup. More synchronized receivers can be employed to generate a high-resolution digital elevation model (DEM) of the observation scene by setting a proper spatial baseline. Moreover, since both the right-hand and left-hand circular polarizations have been employed by the satellite digital TV broadcasting in Japan, a PB-GB-SAR system that can measure different polarizations should be employed to have a better understanding of the imaging scene, like that in GB-SAR [19–21]. Furthermore, in the derivation of the displacement estimation formula in Equation (35), the assumption of "the target only has the displacement along the LoS direction of the surveillance antenna" has been used. If this assumption is not satisfied, only the two-way displacement can be estimated, given by

$$D = \delta y + \delta r = \delta y (1 + \cos \theta) + \delta x \sin \theta \tag{42}$$

In such a case, with a single surveillance antenna, the target displacement along the $x$ and $y$ directions cannot be derived.

To solve this problem, as indicated by Figure 21, two surveillance antennas can be used to get the 2D displacement vector of the target as

$$\begin{cases} \delta x = \frac{D_2(1+\cos\theta_1)-D_1(1+\cos\theta_2)}{\sin\theta_2-\sin\theta_1+\sin(\theta_2-\theta_1)} \\ \delta y = \frac{D_1\sin\theta_2-D_2\sin\theta_1}{\sin\theta_2-\sin\theta_1+\sin(\theta_2-\theta_1)} \end{cases} \tag{43}$$

where $D_a$ ($a$ = 1, 2) is the two-way displacement estimated by the $a$-th surveillance antenna and $\theta_a$ is equal to arctan $[(x - x_0^a)/y]$ with $x_0^a$ being the center position of the $a$-th synthetic aperture.

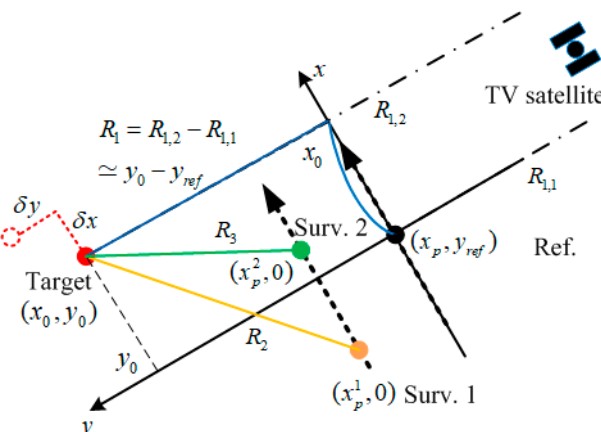

**Figure 21.** Displacement vector estimation by PB-GB-SAR with two surveillance antennas.

At last, it should be pointed out that, although it is assumed to be stationary, the TV satellite indeed has some movements [58]. In the current study with the focus on imaging a local area during a short-term measurement, it has been found that the displacement estimation error is small and thus the phase error caused by the movement of the TV satellite has not been considered. When a large observation scene is to be imaged and long-term measurements are considered, the plane wave approximation may not be suitable and the TV satellite position should be determined in advance to get better imaging and displacement estimation results.

## 6. Conclusions

A passive bistatic ground-based synthetic aperture radar (PB-GB-SAR) system using the geostationary satellite digital TV signal instead of a dedicated transmitter has been designed and validated in this study for high-resolution SAR imaging and high-accuracy displacement estimation purposes. Link budget analysis has proved the applicability of PB-GB-SAR and provided the fundamentals for system parameter selection. The proposed LNB frequency/phase synchronization method can improve the coherence between the reference and surveillance channels. The back projection algorithm and the range migration algorithm can generate focused SAR images. The displacement of the target can be effectively estimated. Some discussions indicate the improvement directions of PB-GB-SAR. We think the developed PB-GB-SAR system has the potential to be used for monitoring landslide, building, and dams, but needs further study.

**Author Contributions:** W.F. conducted the experiments, analyzed the data, and wrote the manuscript; G.N. provided some suggestions for data processing and reviewed the manuscript; S.W. helped to conduct the experiments; J.-M.F. and G.M. conducted LNB synchronization, J.-M.F. also gave some suggestions and reviewed the manuscript; M.S. guided the research, gave some suggestions, and provided some materials for the experiments.

**Funding:** The Besancon Observatory (France) supported this research activity through an SRO grant. This work was also supported by JSPS Grant-in-Aid for Scientific Research (A) 26249058.

**Conflicts of Interest:** The authors declare no conflict of interest.

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
