# Peer review of "Passive Bistatic Ground-Based Synthetic Aperture Radar: Concept, System, and Experiment Results"

_remotesensing, doi:10.3390/rs11151753_

Round 1

Reviewer 1 Report

Thank you for working on "Passive bistatic ground based synthetic aperture radar", with proper investigation and experimentation. It is an interesting work and falls within the scope of the journal. I have given my suggestions to the authors:

(1) The introduction can be enriched with further state of the art references in the field. i have some suggestion that at start discuss regarding antennas for different radars including (a) Sensors 201818(9), 3155 (b) Scientific Reportsvolume 8, Article number: 13367 (2018

(2) Line 47 to 54, The sentence is very lengthy, please split it for proper understanding and rephrase the question properly.

(3) Please provide references for equation (1) to (7). If the manuscript has self-containedFig then also indicate it clearly.

(4) Figure 6: Caption is too lengthy. Can it be minimized by shifting text to the main body?

(5) LNB synchronization section is interesting and helpful for the reader. Please if possible improve the quality of Fig 11 as well.

(6) any reason for utilizing SRRC filter? why not RRC. Also, reference SRRC filter equation from basic filter book.

(7) The artifact suppression section needs to be explained further, as it is difficult to understand from the current text.

Author Response

Thank you very much for your comments, according to which we have carefully modified the manuscript. Please see the attached PDF file.

Reviewer 2 Report

This paper introduces a passive bistatic ground-based SAR system for local area target imaging and displacement estimation applications. The system development has been detailed and some specific signal processing methods have been proposed. Two field experiments have verified the developed radar system and proposed methods. The following are some questions and suggestions on this manuscript:

- The authors have organized their manuscript based on the description of GB-SAR, while there are various PBR systems that use different IO signals for target imaging applications. I recommend the authors make a more detailed introduction of PBR.  

- In the introduction part, the authors have stated that the property of the geostationary satellite digital TV is promising for PBR applications. It is better to make this point clearer. What characteristics are used to evaluate the property of satellite digital TV?

- Since there is thermal noise in the reference channel, the integration gain generated by the cross-correlation process will be decreased. Therefore, the link budget analysis in (6) and (7) may not be suitable. The integration gain loss should be considered.

- In Section 3, the authors have modified two typical imaging algorithms for their developed PB-GB-SAR system. Although the performance of these two algorithms has been validated by experiments, they are not suitable for imaging a large observation area with fine grid size. How the problem of high computational cost can be solved?

- For the displacement estimation by PB-GB-SAR, as mentioned by the authors, there is no single LoS direction. The assumption of “the target only has the displacement along the LoS direction” used to derive (35) is difficult to understand, which should be clarified. When this assumption is not satisfied, how the displacement of the target can be calculated?

Author Response

(The authors gave the same response as above.)

Round 2

Reviewer 1 Report

My comments are addressed and I am satisfied with the current version.